# Kokumi Taste Active Peptides Modulate Salt and Umami Taste

**DOI:** 10.3390/nu12041198

**Published:** 2020-04-24

**Authors:** Mee-Ra Rhyu, Ah-Young Song, Eun-Young Kim, Hee-Jin Son, Yiseul Kim, Shobha Mummalaneni, Jie Qian, John R. Grider, Vijay Lyall

**Affiliations:** 1Korea Food Research Institute, Wanju-gun, Jeollabuk-do 55365, Korea; ahsong47@naver.com (A.-Y.S.); keykos@naver.com (E.-Y.K.); thsgmlwls77@naver.com (H.-J.S.); kimys@kfri.re.kr (Y.K.); 2Department of Physiology and Biophysics, Virginia Commonwealth University, Richmond, VA 23298, USA; shobha.mummalaneni@vcuhealth.org (S.M.); jie.qian@vcuhealth.org (J.Q.); john.grider@vcuhealth.org (J.R.G.)

**Keywords:** Korean soy sauce, kokumi, umami, salty, chorda tympani, amiloride-insensitive salt taste pathway

## Abstract

Kokumi taste substances exemplified by γ-glutamyl peptides and Maillard Peptides modulate salt and umami tastes. However, the underlying mechanism for their action has not been delineated. Here, we investigated the effects of a kokumi taste active and inactive peptide fraction (500–10,000 Da) isolated from mature (FII_m_) and immature (FII_im_) Ganjang, a typical Korean soy sauce, on salt and umami taste responses in humans and rodents. Only FII_m_ (0.1–1.0%) produced a biphasic effect in rat chorda tympani (CT) taste nerve responses to lingual stimulation with 100 mM NaCl + 5 μM benzamil, a specific epithelial Na^+^ channel blocker. Both elevated temperature (42 °C) and FII_m_ produced synergistic effects on the NaCl + benzamil CT response. At 0.5% FII_m_ produced the maximum increase in rat CT response to NaCl + benzamil, and enhanced salt taste intensity in human subjects. At 2.5% FII_m_ enhanced rat CT response to glutamate that was equivalent to the enhancement observed with 1 mM IMP. In human subjects, 0.3% FII_m_ produced enhancement of umami taste. These results suggest that FII_m_ modulates amiloride-insensitive salt taste and umami taste at different concentration ranges in rats and humans.

## 1. Introduction

Mammals use G-protein-coupled receptors (GPCRs) expressed in Type II taste receptor cells (TRCs) to detect bitter, sweet, and umami taste stimuli. While amiloride-sensitive salt taste is detected by Type 1 cells expressing epithelial Na^+^ channels, Type II and Type III cells mediate amiloride-insensitive salt taste. Otopetrin-1 proton selective channel expressed in Type III TRCs detects sour taste stimuli [1,2,3,4,5,6]. Much progress has been made in the identification of taste receptors and the downstream signalling mechanisms involved in the transduction of salty, sour, sweet, bitter and umami taste qualities. However, psychophysical, neural, and cellular studies have long suggested that cell to cell interactions within taste buds and interactions between different taste receptors enhance or suppress taste responses [7,8]. The synergism between monosodium glutamate (MSG) and 5’-ribonucleotides, a distinct characteristic of umami taste, is an example of a binary taste interaction between agonists [9,10]. Additionally, umami peptides modulate bitterness by interfering with ligand binding to the human bitter taste receptor TAS2R16 [11]. Interactions between non-tastants and tastants can also modulate taste intensity. SE-1, a sweet receptor positive allosteric modulator, binds to the sweet receptor without activating it, but does so in a manner that causes the orthogonal ligands to bind with higher affinity [12,13]. 

Kokumi taste has the characteristics of enhancing continuity, thickness, and mouthfeel, and was first observed in an aqueous extract of garlic in an umami solution. Kokumi produces its effect despite minimally eliciting any taste on its own [14]. Sulfur-containing compounds and their γ-glutamyl peptides, including γ-Glu-Cys-Gly (GSH) were suggested to be kokumi-active substances [14,15,16,17]. Because GSH was identified as an endogenous modulator of the calcium-sensing receptor, which participates in calcium homeostasis in the body [18], identification of GSH as an active component suggests the involvement of calcium-sensing receptor in kokumi perception [19]. Subsequent sensory analyses using various extracellular calcium-sensing receptor agonists have shown that kokumi did have a taste-enhancing effect on sweet, salty, and umami taste [19]. Not only does the γ-glutamyl peptide elicit kokumi taste, but the Maillard [20] reacted peptides (MRPs), which are gradually formed by longer maturation of Korean soy sauce, Ganjang (JGN), have been suggested to play a role in the kokumi taste in humans [21]. JGN is generally stored at ambient conditions for a year, and for up to four years or more to attain full maturity. The taste characteristics of kokumi increase as the maturation progresses. 

Salt taste is detected by at least two receptor-mediated pathways. One pathway is Na^+^ specific and involves Na^+^ influx into TRCs that express amiloride- and benzamil (Bz)-sensitive epithelial Na^+^ channels (ENaCs) [22,23]. The second pathway is amiloride-insensitive and is cation nonselective, and does not discriminate between Na^+^, K^+^ and NH_4_^+^ salts. The contribution of these two pathways varies in different taste receptive fields. Approximately 65% of TRCs in the fungiform taste buds exhibit functional ENaCs, 35% of TRCs in foliate taste buds are amiloride-sensitive, while TRCs in the circumvallate are completely amiloride-insensitive, and do not seem to express functional ENaCs [24]. Although at present the identity of the amiloride-insensitive Na^+^ pathways in TRCs remains elusive, the amiloride- and Bz-insensitive salt taste receptors are the predominant transducers of salt taste in humans [25,26,27].

Investigations conducted using the Maillard reaction between peptides (1000–5000 Da) isolated from soy protein hydrolysate and xylose (Xyl-MRPs) have been known to enhance umami, continuity, and mouthfeel in umami solution, support the notion that MRPs are another class of kokumi substances [28]. Interestingly, Xyl-MRPs not only modulate umami taste, but also modulate salt taste. The effect of Xyl-MRPs on salt taste is observed at much lower concentrations than those that increase the umami taste [27]. Over a range of concentrations, Xyl-MRPs [27,29] reversibly enhanced the Bz-insensitive NaCl chorda tympani (CT) taste nerve response in rodents, whereas, at high concentrations, they inhibited the Bz-insensitive NaCl CT response. The effect of Xyl-MRPs on the Bz-insensitive NaCl CT responses were transient receptor potential vanilloid 1 (TRPV1)-dependent. In human sensory evaluation, at low salt concentrations, galacturonic acid MRPs (GalA-MRPs) [27] enhanced human salt taste perception. These data suggest that, in both rodents and humans, MRPs induce changes in amiloride-insensitive salt taste and umami taste. 

In this paper, we investigated the effects of a naturally occurring MRPs fraction (500–10,000 Da, FII) isolated from mature (FII_m_; 4-year old) and immature (FII_im_; 1-year old) JGN on salty and umami taste responses in rodents and human subjects. Effects of FII_m_ and FII_im_ were investigated on the Bz-insensitive NaCl CT responses and their interactions with TRPV1 modulators, and glutamate CT responses in rats. Effects of FII_m_ and FII_im_ were investigated on behavioral responses to NaCl in C57BL/6 mice, and on the sensory evaluation of salty and umami tastes in human subjects. Our results demonstrate that FII_m_ produces concentration-dependent biphasic effects on amiloride-insensitive neural and behavioral responses to NaCl in rodents. Above the concentrations that modulate salty taste, FII_m_ enhanced CT responses to glutamate. In human subjects, FII_m_ produced concentration-dependent biphasic effects on salt taste perception and at higher concentrations enhanced umami taste. These results suggest that FII_m_ modulates salty and umami taste in rodents and humans via similar mechanisms.

## 2. Materials and Methods 

### 2.1. Isolation of FII_m_ and FII_im_ from JGN

FII fraction containing MRPs of molecular weight (MW) ranging between 500 and 10,000 Da was isolated from immature (FII_im_; 1-year old) and mature (FII_m_; 4-year old) JGN with an ultra-filtration unit (Model 840, Amicon Inc., Beverly MA, USA) using YM-10 (MW cutoff 10,000 Da) and YC-05 (MW cutoff 500 Da) membranes (Millipore Co., Bedford, MA, USA) at 2–4 °C under N_2_ pressure. Each fraction was lyophilized and stored in a desiccated freezer at −20 °C until use. FII_m_ was further separated using YM5, YM3 or YM1 Millipore membranes that had a cut off MW of 5000, 3000 and 1000 Da, respectively. These fractions were: FII_m__a_ (MW 500–1000 Da), FII_m__b_ (MW 1000–3000 Da), FII_m__c_ (MW 3000–5000 Da) and FII_m__d_ (MW 5000–10,000 Da). FII_m_ and FII_im_ are the unfractionated MRPs and FII_m(a-d)_ are the sub-fractions of different molecular weight. Successive column chromatography was performed with FII_m_ to obtain aromatic, basic, acidic, and neutral conjugated peptide fractions using activated charcoal (60 cm long and 4.0 cm I.D.; Junsei Chemical Co. Ltd., Tokyo, Japan), cation-exchanger (60 cm long and 3.0 cm I.D.; Amberlite IRC-50), and anion-exchanger (60 cm long and 3.0 cm I.D.; Amberlite IRA 400, both from Sigma Co. Ltd., St. Louis, MO, USA) [30,31].

### 2.2. CT Taste Nerve Recordings

In contrast to glossopharyngeal nerve response to NaCl, the predominant NaCl CT response in rodents is amiloride sensitive. However, a significant part of the NaCl CT response is Bz- and amiloride insensitive across the concentration-response range of NaCl [32]. The identity of the amiloride-insensitive receptor presently at best remains elusive in the circumvallate taste receptive field. Our previous studies suggest that in the anterior tongue the amiloride-insensitive pathway is a non-selective cation channel that is sensitive to resiniferatoxin (RTX), N-(3-methoxyphenyl)-4-chlorocinnamide, SB-366791 (SB), capsazepine, iodo-RTX, and temperature [33,34]. We have previously investigated the effect of various salt taste modulators on the Bz-insensitive NaCl CT response using both rats and mice [27,29,32,33,34,35,36]. To compare the results of the effects of FII_m_ on neural responses to NaCl with previously published results, we monitored Bz-insensitive NaCl CT response in rats.

Animals were housed in the Virginia Commonwealth University (VCU) animal facility in accordance with institutional guidelines. All animal protocols were approved by the Institutional Animal Care and Use Committee (IACUC #AD20116). Female Sprague-Dawley rats (150–200 gm) were anesthetized by intraperitoneal injection of pentobarbital (60 mg/Kg) and supplemental pentobarbital (20 mg/Kg) was administered as necessary to maintain surgical anesthesia. The animal’s corneal reflex and toe-pinch reflex were used to monitor the depth of surgical anesthesia. Body temperatures were maintained at 37 °C with a Deltaphase Isothermal PAD (Model 39 DP: Braintree Scientific Inc. Braintree, MA, USA). The left CT nerve was exposed laterally as it exited the tympanic bulla and placed onto a 32G platinum/iridium wire electrode. CT responses were recorded and analyzed as described previously [27,29,32,33,34,35,36]. 

The composition of rinse and NaCl stimulating solutions is shown in Table 1. CT responses in rats were monitored while the anterior lingual surface was stimulated first with the rinse solution (R) and then with salt solutions containing 0.1–0.5% FII_im_, FII_m_ and FII_m_ sub-fractions (FII_m(a–d)_). The pH of the rinse solution and the salt solutions was adjusted to 6.1. In some experiments Bz (5 μM) was added to salt solutions to block Na^+^ entry into TRCs through apical epithelial Bz-sensitive ENaCs. CT responses were also recorded at 23 °C and 42 °C. In additional experiments we tested the effect of FII_m_ on the CT response to MSG and MSG + 5’-inosine monophosphate (IMP), a specific modulator of umami taste [37]. CT responses to MSG were monitored in the presence of Bz to eliminate the contribution of Na^+^ to the glutamate CT response [33] and SB (1 μM), a TRPV1 blocker [38]. In CT experiments the tonic (steady-state) part of the NaCl CT response or glutamate CT response was quantified and normalized to CT responses to 0.3M NH_4_Cl. The normalized data were reported as the mean (M) ± SEM of the number of animals (*n*). Student’s t-test was employed to analyze the differences between sets of data. Since we are comparing the normalized CT responses to NaCl + Bz before and after FII_im_, FII_m_ or FII_m_ sub-fractions in the same CT preparation, paired t-test was used to evaluate statistical significance.

### 2.3. Behavior Studies in Mice 

Rats have a high preference for NaCl even in the presence of Bz [29]. Because of already high background NaCl preference, small increases in NaCl preference are difficult to evaluate in rats. In contrast, mice demonstrate a more moderate preference for NaCl and small shifts in the preference curve are easily detected. Therefore, mice were used for behavioral studies. Behavioral studies were performed in WT (C57BL/6J) mice (30–40 gm) using standard two bottle/48 h tests [39]. Both males and females were used. The care and use of the mice followed the institutional and national guidelines, and the protocol was approved by the committee on the Ethics of Animal Experiments of the Korea Food Research Institute (Permit Number: KFRI-M-12028). Mice (63–70 days of age) were housed in separated cages and were maintained on a standard laboratory chow (Pico-Lab Rodent Diet 20–5053, PMI Feeds) and water *ad libitum*. The air-conditioned animal room was maintained at 22 ± 2 °C, with relative humidity of 59 ± 1% and a 12 h light/dark cycle (light period, 07:00–19:00 h). Each mouse was tested at approximately the same time of day. Before the start of the experiment mice were given two bottles with water for 2 weeks. The experiment was started when mice were accustomed to drinking equally from 2 bottles. Mice were given a choice between two bottles, one containing water and the other a test solution in the following order: water, 30 mM NaCl, 80 mM NaCl, 100 mM NaCl, 120 mM NaCl, 150 mM NaCl, 200 mM NaCl and 300 mM NaCl. We also performed behavioral studies when both water and the NaCl solutions contained 10 µM amiloride. In some experiments, mice were given a choice between water and 100 mM NaCl or between water + 10 µM amiloride and 100 mM NaCl + 10 µM amiloride containing varying concentrations (0.1, 0.25, 0.5, 0.75, and 1.0%) of FII_m_. For each 48 h period the mass of water versus the mass of the test solution consumed by each mouse/g BW was measured. The preference ratio for a taste stimulus was calculated as the mass of the test solution consumed/48 h/g BW divided by the mass of the total fluid intake (mass water + mass of the test solution)/48 h/g BW). The bottles containing water or the test solution were switched from left to right every day. The data were analyzed using one-sample t-tests against 0.5, a reference value meaning indifference of the test solution with respect to the control solution.

### 2.4. Human Sensory Evaluation

All human sensory evaluation protocols were approved by the Public Institutional Review Board Designated by Ministry of Health and Welfare, South Korea. The ethic approval code is P01-202004-23-004. Each participant signed a consent form to participate in salt taste sensory evaluations. To maintain a subject’s confidentiality, the personal data was coded and the taste data were analyzed off line. Previously trained panelists (men and women, ages between 25 to 37 years) with no history of basic taste disorders were recruited. The panelists washed their mouth after tasting each samples. The data was analyzed by one-way ANOVA to compare between-group differences. 

#### 2.4.1. Salt Sensory Evaluation

Panelists were trained to recognize salt taste intensity with reference to 0.2%, 0.35%, 0.5%, and 0.7% NaCl solution representing a value of 2.5, 5.0, 8.5, and 15.0, respectively, using a 15-point intensity scale [40]. To evaluate the effect of FII_im_ and FII_m_ on salt taste, FII_im_ or FII_m_ (0–0.01%) dissolved in 0.2% NaCl solution was presented to the panelists and the salt taste intensity was evaluated with reference to 0.2% NaCl (intensity scale value = 2.5; R1), and 0.35% NaCl (intensity scale value = 5.0; R2), respectively. 

#### 2.4.2. Umami Sensory Evaluation

According to the manufacturer’s instructions, Japanese fish soup base, Hondashi (0.04 g) was dissolved in 100 ml water. The 0.04% Hondashi fish soup base was used as a control and was given an intensity of 5 on a 10-point intensity scale. FII_m_ at 0.003%, 0.01%, 0.03% and 0.3% was dissolved in 0.04% Hondashi Fish soup base and their effect was evaluated on umami taste by the same trained panelists (*n* = 6). As a control, FII_m_ was dissolved in water at 0.003%, 0.01%, 0.03% and 0.3%. These concentrations of FII_m_ were evaluated for umami taste by the same panelists (*n* = 6).

## 3. Results and Discussion

### 3.1. Effect of FII_m_ and FII_im_ on the Bz-insensitive NaCl CT Response

As shown in a representative CT trace (Figure 1A), adding increasing concentrations of FII_m_ to 100 mM NaCl + 5 µM Bz (NaCl + Bz) solution (Table 1) initially produced an increase in both phasic and tonic NaCl CT response of between 0.1% and 0.5%. Above 0.5% FII_m_ the magnitudes of the phasic and tonic CT responses were less than their respective maximum values. In the presence of 1% FII_m_, the tonic CT response decreased below the NaCl + Bz CT response in the absence of FII_m_ (Figure 1A). The variation of the normalized mean tonic NaCl + Bz CT response plotted as a function of the log of FII_m_ or FII_im_ concentrations (%) are summarized in Figure 1B. FII_m_ produced a biphasic dose-response relationship for both the phasic (data not shown) and tonic (Figure 1B; ●) NaCl + Bz CT response. The maximum increase in the mean normalized tonic CT response occurred at 0.5% of FII_m_, an 88% increase relative to NaCl + Bz tonic CT response in the absence of FII_m_. At 1% FII_m_, the tonic NaCl + Bz CT response was significantly less than the tonic CT response with NaCl + Bz alone (*p* = 0.0466; *n* = 3). Stimulating the tongue with the rinse solution (R) containing varying concentrations of FII_m_ elicited only transient (phasic) CT responses that were concentration-independent and were indistinguishable from the mechanical rinse artifact (data not shown). These results indicate that, at the concentrations used in these experiments, FII_m_, by itself, is not a gustatory stimulus in the fungiform taste receptive field and only modulate the CT response in the presence of salt (NaCl + Bz). In contrast, FII_im_ did not produce any changes in either the phasic (data not shown) or the tonic CT response between 0.1% and 1% (Figure 1B; ○). The Xyl-MRPs, GalA-MRPs, glucosamine-MRPs, and fructose-MRPs also produced biphasic effects on the Bz-insensitive NaCl CT response with maximum increase at 0.25%, 0.25%, 0.50%, 0.75% and 1%, respectively [27]. This suggests that both MRPs naturally generated during longer maturation and synthesized in vitro have a common property of producing biphasic effects on the Bz-insensitive NaCl CT response. The potency of MRPs depends upon the reacted sugar moiety. However, at present it is not known which sugar moieties are conjugated to the peptides comprising FII_m_. FII_m_ is a mixture of MRPs of varying molecular weights, charge and affinity for their putative salt taste receptor(s). In comparison, 0.27% GalA-MRPs enhanced the Bz-insensitive NaCl CT response by 101% [27], suggesting naturally occurring FII_m_ produces effects on the NaCl + Bz CT response that are comparable to those produced by the GalA-MRPs.

### 3.2. Effect of SB and FII_m_ on the Bz-insensitive NaCl CT Response 

In our previous studies, Bz-insensitive NaCl CT responses in rodents were inhibited by TRPV1 blockers. In addition, Bz-insensitive NaCl CT responses were not observed in TRPV1 knockout mice [33]. Accordingly, in the next series of experiments we tested if FII_m_ effects on salt responses were also sensitive to SB, a TRPV1 blocker. Because 0.4% and 0.5% FII_m_ give almost equivalent CT responses (Figure 1B), we used 0.4% FII_m_ in these experiments. In mixtures containing NaCl + Bz + SB, the constitutive NaCl + Bz tonic CT response was inhibited to the rinse baseline level (Figure 2A,C). Subsequently, stimulating the rat tongue with solutions containing NaCl + Bz + SB + 0.4% FII_m_ significantly inhibited the CT nerve response relative to NaCl + Bz + 0.4% FII_m_ (Figure 2A,C; ** *p* = 0.0001; *n* = 3). These results suggest that both the constitutive amiloride- and Bz-insensitive CT response and the subsequent FII_m_-induced increase in the CT response are SB-sensitive. 

### 3.3. Effect of RTX and FII_m_ on the NaCl + Bz CT Response

Consistent with previous studies [33,35], at room temperature (23 °C), RTX (0.25 μM) enhanced the rat NaCl + Bz CT response relative to NaCl + Bz (Figure 2A,C; * *p* = 0.0001; *n* = 3). When the tongue was stimulated with NaCl + Bz solutions containing both RTX (0.25 μM) and FII_m_ (0.4%), no further increase in the magnitude of the Bz-insensitive NaCl CT response was observed relative to NaCl + Bz + RTX (Figure 2C). These results suggest that RTX and FII_m_ target the same amiloride-insensitive pathway(s). The Bz-insensitive NaCl taste responses are regulated by several intracellular signaling mediators. A decrease in taste cell Ca^2+^, activation of protein kinase C, and inhibition of calcineurin enhanced the magnitudes of the Bz-insensitive NaCl CT responses in the presence of RTX, and either minimized or completely eliminated the decrease in the CT response at RTX concentrations >1 µM. In contrast, increasing taste cell Ca^2+^ inhibited the Bz-insensitive NaCl CT response in the presence of RTX [41]. An increase in taste cell phosphatidylinositol 4,5-bisphosphate inhibited the control NaCl + Bz CT response and decreased its sensitivity to RTX. Alternately, a decrease in phosphatidylinositol 4,5-bisphosphate enhanced the control NaCl + Bz CT response, increased its sensitivity to RTX stimulation, and inhibited the desensitization of the CT response at RTX concentrations >1 µM [42]. It is likely that Bz-insensitive NaCl CT responses in the presence of FII_m_ are also regulated by the above intracellular modulators and are responsible for their biphasic effects on the NaCl CT response.

### 3.4. Effect of Elevated Temperature and FII_m_ on the NaCl + Bz CT Response 

In our previous studies, Bz-insensitive NaCl CT responses in rodents were temperature dependent. In addition, temperature and modulators of the Bz-insensitive NaCl CT response produced additive effects on CT response [26,33,36]. Accordingly, we next tested the effect of elevating the temperature from 23 °C to 42 °C on the CT response to NaCl + Bz and NaCl + Bz + 0.4% FII_m_. As shown in a representative CT recording, elevating the temperature from 23 °C to 42 °C increased the magnitude of the tonic NaCl + Bz CT response relative to 23 °C (Figure 2B). FII_m_ (0.4%) further increased the CT response at 23 °C and 42 °C (Figure 2B). The mean tonic NaCl + Bz CT response at 23 °C (Figure 2C) was significantly enhanced by increasing the temperature to 42 °C (* *p* = 0.0039) and by the addition of 0.4% FII_m_ (Figure 2C; ** *p* = 0.0001; *n* = 3). These results show that elevated temperature and FII_m_ produce additive effects on the amiloride-insensitive NaCl CT response.

### 3.5. Effect of FII_m_ sub-fractions of Different Molecular Weights (FII_m(a-d)_) on the NaCl + Bz CT Response 

FII_m_ was further separated into four sub-fractions of varying molecular weights: FII_ma_ (500–1000 Da), FII_mb_ (1000–3000 Da), FII_mc_ (3000–5000 Da) and FII_md_ (5000–10,000 Da). As shown in representative CT recordings in Figs. 3A and 3B, the relationship between varying concentrations of FII_ma_ and FII_mc_ and the magnitude of NaCl + Bz CT response was shifted to the right on the concentration axis relative to FII_m_ (Figure 1A). The relationships between varying concentrations of FII_ma_, FII_mb_, FII_mc_ and FII_md_ and the corresponding mean normalized tonic NaCl + Bz CT response are plotted in Figure 3C. The results show that for all sub-fractions FII_m(a–d)_, the relationship between their concentrations and the magnitude of tonic NaCl + Bz CT response is shifted to the right on the concentration axis relative to FII_m_. FII_ma_ produced the maximum increase in the NaCl + Bz tonic CT response at a concentration between 1.5 and 2.5% (Figure 3C; ◆). This concentration is significantly higher than the concentration at which FII_m_ produced the maximum increase in the NaCl + Bz CT response (0.5%; Figure 1B). These results suggest that FII_m_ fraction is composed of MRPs of varying molecular weights that differ in their affinity and potency in modulating the putative amiloride-insensitive salt taste pathway(s). 

### 3.6. Effect of FII_m_ sub-fractions (Neutral, Acidic, Basic and Aromatic) on the NaCl + Bz CT Response 

FII_m_ was further separated into neutral, acidic, basic and aromatic sub-fractions. Since the relationships between varying concentrations of the neutral, acidic and basic fractions and the magnitude of the tonic NaCl + Bz CT response were very similar in individual rats, the data from these three fractions were combined and are plotted in Figure 4A (▲). In all three fractions, the relationship between their concentrations and the magnitude of the mean normalized tonic NaCl CT response was shifted to the right on the concentration axis relative to FII_m_ (Figure 4A; ●). In contrast, the aromatic fraction produced a biphasic response in the NaCl + Bz CT response with a very sharp-peak at 0.75% (Figure 4A; ○). These results further suggest that FII_m_ is composed of neutral, acidic, basic and aromatic MRPs that show varying degrees of potency and affinity for modulating the putative amiloride-insensitive salt taste pathway(s). It is interesting to note that relative to control (NaCl + Bz), 0.5% FII_m_ (Figure 1A) produced an equivalent maximum increase in the tonic NaCl + Bz CT response as 1 µM RTX (Figure 4B).

We also recorded FII_m_-induced changes in the Bz-insensitive NaCl CT response in wild type (WT; C57BL/6J) and homozygous TRPV1 knockout mice (B6. 129S4-Trpv1^tmijul^; Jackson Laboratory, Bar Harbor, ME). Consistent with our earlier study with MRPs [27], FII_m_ produced a similar biphasic response on the Bz-insensitive NaCl CT response in WT mice. In TRPV1 KO mice, FII_m_ (0.4%) did not induce CT response above the rinse baseline (data not shown). This is akin to our results in rats. SB inhibited the basal Bz-insensitive NaCl CT response. In the continuous presence of SB, FII_m_ produced a significantly smaller increase in the Bz-insensitive NaCl CT response relative to the absence of SB (Figure 2). These results indicate that FII_m_ produces similar effects on rats and mice.

### 3.7. Effect of Calcitonin Gene Related Peptide (CGRP) on NaCl CT Responses 

RTX activates and SB inhibits amiloride-insensitive NaCl CT responses (Figure 2). However, TRPV1 immunoreactivity was not found in TRCs [43,44,45]. We hypothesize that RTX and other modulators of TRPV1 alter Bz-insensitive NaCl CT responses indirectly, by releasing CGRP from trigeminal nerves [46]. The released CGRP then acts on its specific receptor (CGRPR) in TRCs to modulate Bz-insensitive NaCl CT responses [47].

Due to the concern that topical lingual application of CGRP, a large neuropeptide, may not be able to reach its receptor in TRCs, CGRP was administered by intraperitoneal injection. CT responses were monitored while the rat tongue was stimulated with 0.3M NH_4_Cl, 0.3 M NaCl and 0.1M NaCl before and after an i.p. injection of 23 µg/100 BW or 68 μg/100 g BW CGRP dissolved in 0.5 mL PBS. Following i.p. injection of 68 μg/100 g BW CGRP the NaCl CT response increased with time (data not shown). As shown in Figure 5A, 10 min post CGRP injection, the NaCl CT responses were almost two times greater than control (Figure 5B). CGRP also induced an increase in the CT response to 0.3M NaCl. However, an i.p. injection of 23 µg/100 BW CGRP did not induce any changes in rat NaCl CT response 10 min post CGRP injection (Figure 5B). These results indicate that CGRP effects on NaCl CT response are both time- and dose-dependent and are observed over a range of NaCl concentrations. These results suggest a possible interaction between the trigeminal and salt taste systems. 

Using calcium imaging, a subset of acid responsive Type III mouse circumvallate TRCs were identified as the amiloride-insensitive salt responsive cells [2]. CGRPR has been suggested as the functional link to cellular transduction pathway for CGRP action on Type III TRCs. CGRP has been shown to increase [Ca^2+^] in Type III TRCs. This effect of CGRP was dependent upon phospholipase C activation and was prevented by U73122 [47]. In mouse taste buds, CGRP caused TRCs to secrete serotonin (5-HT), a presynaptic (Type III) cell transmitter. 5-HT seems to reduce taste evoked ATP secretion in Type II cells [47]. However, at present this information is lacking in the fungiform taste receptive field.

Here, we present new data that suggest that CGRP can modulate rat amiloride-insensitive NaCl CT responses (Figure 5). In a recent study [3] amiloride-insensitive Ca^2+^ responses in mouse taste bud cells were localized to the apical tips of Type II, but not in Type III fungiform TRCs. It is suggested that, because anterior (fungiform) and posterior (circumvallate) taste fields differ functionally, in an earlier study [2] amiloride-insensitive NaCl responses may have been detected in only Type III circumvallate taste cells. Although the identity and location of the amiloride- and Bz-insensitive pathway(s) are ambiguous at present, CT recordings demonstrate that a component of the amiloride- and Bz-insensitive NaCl CT response at low NaCl concentrations (100 mM) is present in the anterior taste field that is modulated by RTX, FII_m_, temperature, SB (Figure 1A and Figure 2B) and voltage [33,34]. 

*N*-geranyl cyclopropylcarboxamide (NGCC), a modulator of the amiloride- and Bz-insensitive NaCl CT responses [48], activates hTRPV1 expressed in HEK293T cells [49]. In our preliminary studies, component of FII_m_ induced inward current in TRPV1-expressing cells in whole-cell patch-clamp recordings [50,51]. Currently studies are underway to demonstrate direct activation of the expressed umami taste receptor by FII_m_. However, at present it is not clear if, like RTX, other modulators of the amiloride-insensitive pathway release CGRP from TRPV1 expressed in trigeminal neurons in a dose-dependent manner. In addition, it is also not known if, like other modulators of the amiloride-insensitive NaCl CT response, CGRP elicits a biphasic effect on rat NaCl CT responses. Taken together, our data suggest a possible linkage between the trigeminal system and amiloride-insensitive salt taste. It has recently been demonstrated that sour taste pathway works together with the somatosensory system to trigger aversive responses to acidic stimuli [5]. 

### 3.8. Behavioral Studies with Mice

Under control conditions, mice demonstrated a bell shaped NaCl preference curve with a significant preference for 100 mM NaCl (Figure 6A; ○; * *p* = 0.02; *n* = 10) and aversion for 300 mM NaCl (**** *p* = 0.0001) [39]. In the presence of 10 µM amiloride the NaCl preference curve was again biphasic but was shifted to the right on the NaCl concentration axis. In the presence of amiloride, mice showed a significant NaCl preference at 150 mM NaCl (Figure 6 A; ●; *** *p* = 0.0024). 

As shown in Figure 6B, adding increasing concentrations of FII_m_ (0.1 to 1%) to 100 mM NaCl solutions in the absence and presence of 10 µM amiloride produced biphasic changes in NaCl preference, increasing it at 0.25% and lowering it at higher concentrations. Under control conditions, FII_m_ maximally enhanced the NaCl preference at 0.25% relative to NaCl alone (Figure 6B; ○; ** *p* = 0.0001; *n* = 10). Above 0.25% FII_m_ NaCl preference was significantly less than its maximum value. In the presence of 10 µM amiloride the maximum increase in NaCl preference was observed at 0.5% FII_m_ (Figure 6B; ●; * *p* = 0.0086). Above 0.5% FII_m_ NaCl preference was significantly less than its maximum value. There was no change in NaCl preference when equivalent concentrations of the FII_im_ were added to the test solutions containing 100 mM NaCl or 100 mM NaCl + 10 µM amiloride (data not shown). These behavioral responses to NaCl are correlated with the biphasic effects of FII_m_ concentrations on the amiloride- and Bz-insensitive NaCl CT responses (Figure 1). In this sense, FII_m_ mimics the effect of other modulators [27,33,34,36,48,52] of the amiloride- and Bz-insensitive NaCl CT responses.

### 3.9. Effect of FII_m_ on Salt Taste in Human

In human sensory evaluation, FII_m_ produced a biphasic effect on salt taste. FII_m_ increased salt taste intensity between 0.3 and 0.5%, but slightly decreased it above 0.5% (Figure 7). The maximum salt taste intensity in human subjects was detected at 0.5% FII_m_ (Figure 7; ●). In contrast, FII_im_ had no significant effect on human salt taste perception (Figure 7; ○). In our previous studies, GalA-MRPs, Xyl-MRPs [27] and NGCC [48], modulators of the amiloride- and Bz-insensitive NaCl CT responses in rodents, also produced biphasic effects on human salt taste intensity. Although functional ENaC channels are expressed in human fungiform TRCs [53], the amiloride- and Bz-insensitive salt taste receptors are the predominant transducers of salt taste in humans [25,26,27]. Thus, modulation of the amiloride-insensitive salt taste in humans via FII_m_ or other modulators may provide alternate ways to regulated human salt taste and perhaps salt intake.

### 3.10. Effect of FII_m_ on the Rat CT Response to Glutamate 

Stimulating the rat tongue with 100 mM MSG + Bz + SB elicited a CT response and the CT response was enhanced in the presence of 1 mM IMP (Figure 8A). Glutamate CT response was also enhanced in the presence of 2.5% FII_m_. The normalized tonic CT responses to glutamate in the absence and presence of IMP and FII_m_ are summarized in Figure 8B. FII_m_ at 2.5% enhanced the CT response to glutamate that was equivalent to the enhancement observed with 1 mM IMP. These results further suggest that unlike the Bz-insensitive NaCl CT response, the basal umami CT response and the subsequent FII_m_ induced enhancement of the umami CT response is SB-insensitive. In our previous study, Xyl-MRs also enhanced the CT response to glutamate at concentrations above which they modulated the NaCl + Bz CT responses. In contrast to Xyl-MRPs, at these concentrations GalA-MRPs or Glc-NH_2_-MRPs did not show effects on the glutamate CT response [27]. These results suggest that the umami enhancing effect of MRPs is dependent on the conjugated sugar(s). However, at present the identity of the specific sugar resides conjugated with the peptides comprising the FII_m_ is not known.

In our earlier study [33], RTX demonstrated a biphasic response on the rat Bz-insensitive NaCl CT response. At 1 μM, it maximally enhanced and at 10 μM, maximally inhibited the Bz-insensitive NaCl CT response. At 1 and 10 μM concentrations, RTX did not alter CT responses to 500 mM sucrose, 10 mM quinine and 10 mM HCl. These results tend to suggest that over the concentration range that alter the Bz- insensitive NaCl CT response, modulators of the amiloride-insensitive pathway may not alter sweet, bitter or sour taste. At present it is not known if FII_m_ concentrations that modulate salt responses also alter responses to other taste stimuli.

### 3.11. Effect of FII_m_ on Umami Taste in Human

In human subjects, adding 0.3% FII_m_ to umami soup base produced a significant increase in umami taste intensity (Figure 8C; open bars; *p* < 0.01; *n* = 9). Equivalent concentrations of FII_m_ added to water produced umami intensity ratings of < 1 (Figure 8C; filled bars). In contrast to a strong salt taste enhancing effect at 0.5% FII_m_, lower concentrations (<0.3%) of the FII_m_ did not have a significant effect on umami taste intensity. Thus, depending upon the concentration, MRPs can be used either as salt taste or umami taste modifiers.

These results show that FII_m_ modulates both salt and umami taste in humans but at different concentration range. The differences in the sensitivity to FII_m_ between humans and mice are most likely due to the variations in the umami taste receptor protein [54,55]. This dual property of being able to modulate the Bz-insensitive NaCl response and the glutamate response at two different concentration ranges is not restricted to FII_m_. We have recently shown that at different concentrations ranges, NGCC modulates Bz-insensitive salt taste responses and glutamate taste responses in humans and animal models [48]. 

## 4. Conclusions

In summary, a naturally occurring kokumi taste active peptide fraction (MW 500–10,000 Da) isolated from mature (FII_m_; 4-year old) Ganjang, a typical Korean Soy Sauce, modulates the amiloride-, Bz-insensitive NaCl CT response in rodents in a biphasic manner. At low concentrations (0.1 to 0.5%) it enhanced and at higher concentrations (>0.5%) inhibited the Bz-insensitive NaCl CT response. FII_m_ effects on Bz-insensitive NaCl CT responses are TRPV1 dependent. FII_m_ may indirectly alter CT responses to NaCl via the release of CGRP from trigeminal fibers near the fungiform taste buds in the anterior taste field. This suggests a novel relationship between trigeminal system and salt taste perception. At concentrations >1%, FII_m_ enhanced the CT response to glutamate. In human sensory tests, FII_m_ increased the salt taste intensity between 0.3 and 0.5%, and the umami taste intensity at 0.3%. We conclude that, depending upon its concentration, FII_m_ modulates both salty and umami tastes in humans and rodents. The active component(s) and salt enhancing property of naturally occurring MRPs by longer maturation in food should be further investigated for a better understanding of the potential link between the compound and its beneficial effect in reducing salt intake in the human population. 

## Figures and Tables

**Figure 1 nutrients-12-01198-f001:**
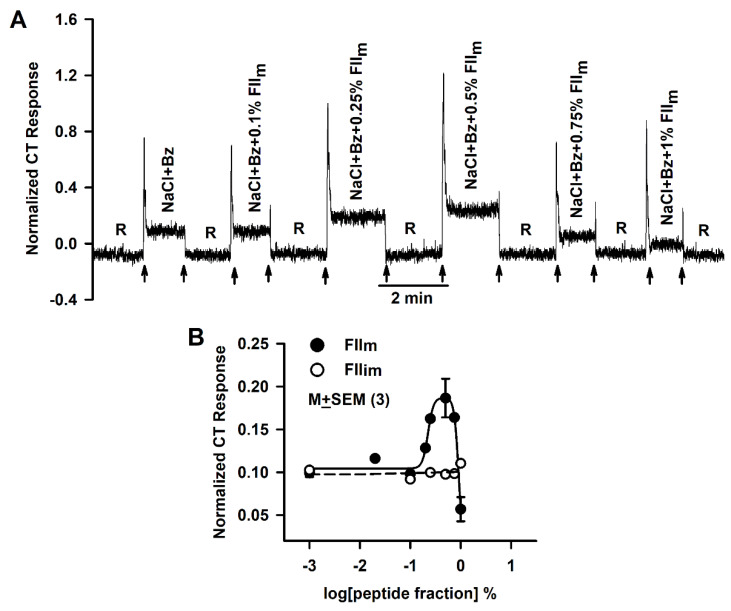
Effect of FII_im_ and FII_m_ on the benzamil (Bz)-insensitive NaCl chorda tympani (CT) response. (**A**) Shows a representative trace in which the CT responses were monitored while the rat tongue was first superfused with a rinse solution (R) and then with a stimulating solution containing 100 mM NaCl + 5 μM Bz + FII_m_ (0–1%) maintained at room temperature. The arrows represent the time periods when the rat tongue was superfused with R and the stimulating solutions. The data were normalized to the tonic response obtained with 0.3 M NH_4_Cl. (**B**) Shows the mean normalized tonic NaCl CT responses in different sets of 3 rats each while their tongues were first stimulated with R and then with NaCl + Bz solutions containing 0–1% of the FII_m_ (●) or FII_im_ (○) expressed in log units. The values are M ± SEM of 3 rats.

**Figure 2 nutrients-12-01198-f002:**
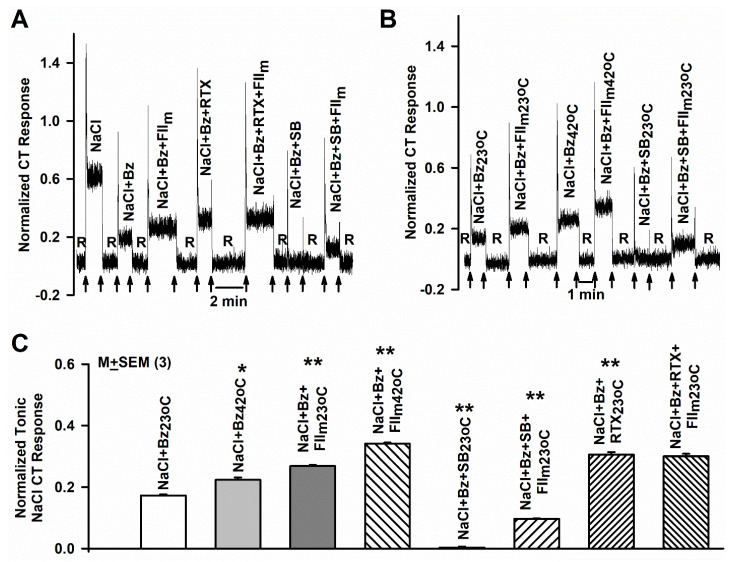
Effect of resiniferatoxin (RTX), SB-377791 (SB), FII_m_ and temperature on the benzamil (Bz)-insensitive NaCl chorda tympani (CT) response. (**A**) Shows a representative CT trace obtained while the rat tongue was first stimulated with rinse solution (R) and then with NaCl, NaCl + Bz, NaCl + Bz + 0.4% FII_m_, NaCl + Bz + 0.25 μM RTX, NaCl + Bz + 0.4% FII_m_ + 0.25 μM RTX, NaCl + Bz + 1 μM SB and NaCl + Bz +SB + 0.4% FII_m_ maintained at room temperature. The data were normalized to the tonic response obtained with 0.3 M NH_4_Cl. The arrows represent the time periods when the rat tongues were superfused with R and the stimulating solutions. (**B**) Shows a representative CT trace obtained while the rat tongue was first stimulated with R at 23 °C (R_23_
_°C_) and then with NaCl + Bz (NaCl + Bz_23_
_°C_), NaCl + Bz + 0.4% FII_m_ at 23 °C (NaCl + Bz + FII_m23_
_°C_), NaCl + Bz at 42 °C (NaCl + Bz_42_
_°C_) and NaCl + Bz + 0.4% FII_m_ at 42 °C (NaCl + Bz + FII_m42_
_°C_). The trace also shows the CT response in the presence of NaCl + Bz + SB and NaCl + Bz + SB + 0.4% FII_m_ maintained at 23 °C. The data were normalized to the tonic response obtained with 0.3 M NH_4_Cl. The arrows represent the time periods when the rat tongues were superfused with R and the stimulating solutions. (**C**) Shows the M ± SEM normalized rat tonic NaCl + Bz CT responses at 23 °C and 42 °C in the absence and presence of 0.4% FII_m_. All unpaired comparisons were made with respect to the normalized value of the tonic CT response to NaCl + Bz at 23 °C. * *p* = 0.0038; ** *p* = 0.0001; *n* = 3).

**Figure 3 nutrients-12-01198-f003:**
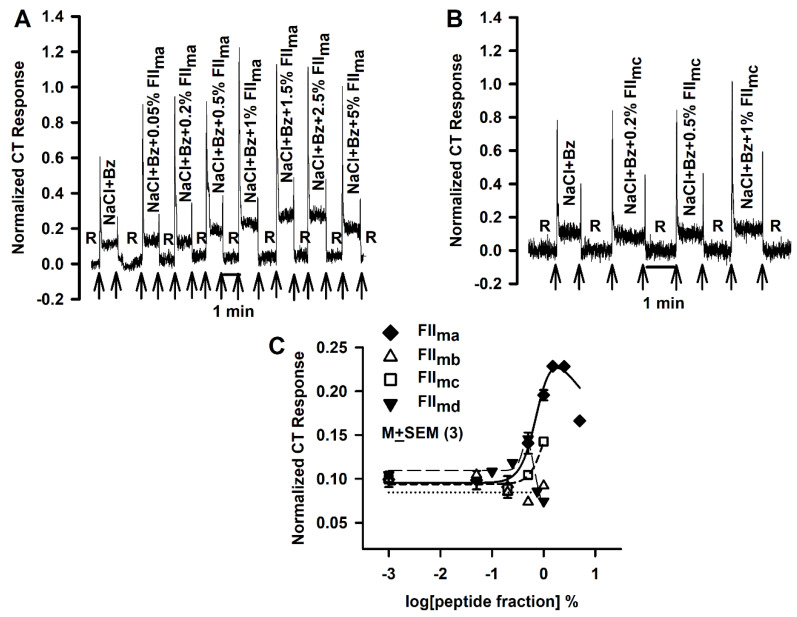
Effects of FII_m_ sub-fractions (FII_m(a-d)_) on the benzamil (Bz)-insensitive NaCl chorda tympani (CT) response. Representative CT responses showing the effect of adding varying concentrations of FII_m_ sub-fractions FII_ma_ (500–1000 Da) (**A**) and FII_mc_ (1000–3000 Da) (**B**) on the rat CT responses to NaCl + Bz. The arrows represent the time period when the tongue was superfused with the rinse and stimulating solutions. In each rat the data were normalized to the tonic response obtained with 0.3M NH_4_Cl. (**C**) Shows the mean normalized tonic NaCl CT responses in different sets of 3 rats each while their tongues were first stimulated with R and then with NaCl + Bz solutions containing 0–1% of the four FII_m_ sub-fractions in log units. The values are M ± SEM of 3 rats in each group. In each case the data were fitted to Equation (4).

**Figure 4 nutrients-12-01198-f004:**
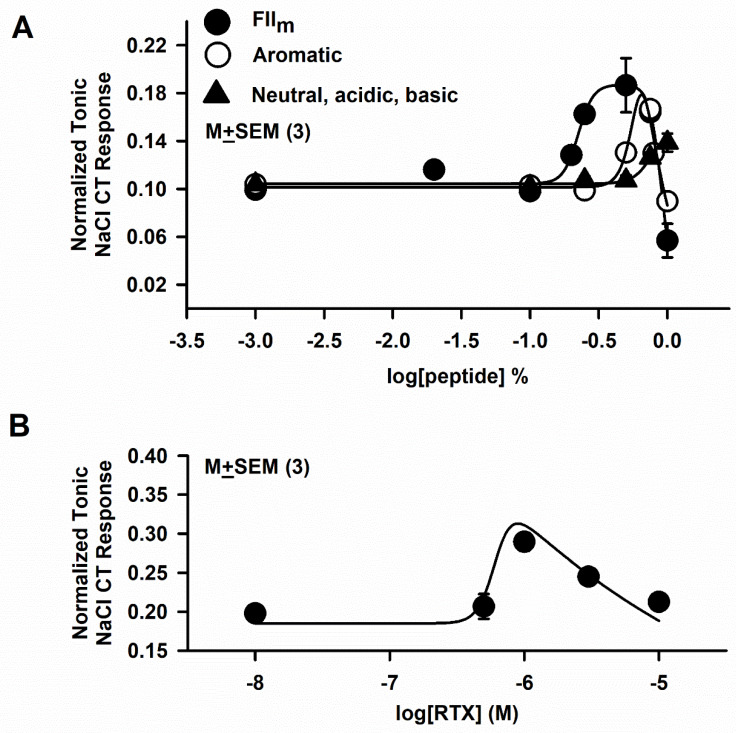
Effects of aromatic, neutral, acidic and basic FII_m_ sub-fractions on the benzamil-insensitive NaCl chorda tympani (CT) response. (**A**) Shows the relationship between varying FII_m_ sub-fraction concentrations expressed in log units and the mean normalized tonic NaCl CT response from 3 rats in each group for FII_m_ (●), aromatic (○) and combined neutral, acidic and basic maillard reacted peptides (▲). (**B**) Shows the relationship between resiniferatoxin (RTX) concentrations expressed in log units and the mean normalized tonic NaCl CT responses from 3 rats (●). The values are M ± SEM of 3 rats in each group.

**Figure 5 nutrients-12-01198-f005:**
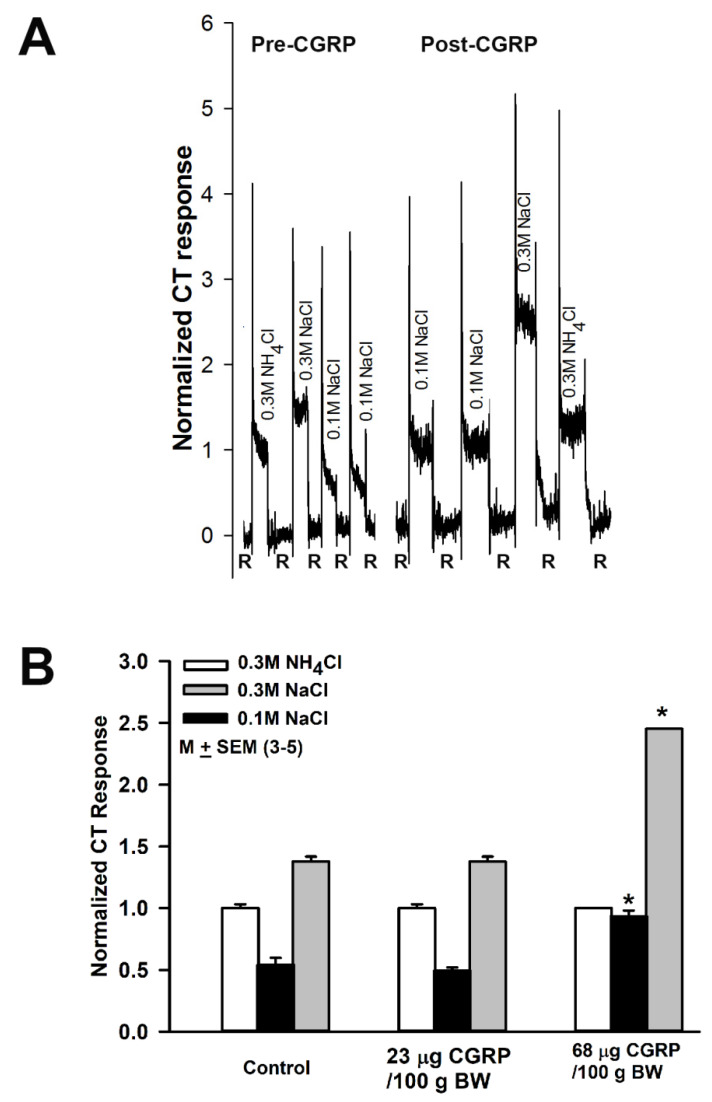
Effect of i.p. injection of calcitonin gene related peptide (CGRP) on NaCl chorda tympani (CT) response. (**A**) Shows a representative CT trace obtained while the rat tongue was first stimulated with rinse solution (R) and then with 0.3M NH_4_Cl, 0.3M NaCl and 0.1M NaCl before and after i.p. injection of CGRP (68 μg/100 g BW in PBS). In each rat the data were normalized to the tonic response obtained with 0.3M NH_4_Cl. The values are M ± SEM of 3 rats in each group. (**B**) Shows summary of the data from 3 rats in each group injected with either 23 or 68 μg CGRP/100 g BW. Values are M ± SEM of 3 rats. * *p* = 0.017 (0.1M NaCl) and 0.009 (0.3M NaCl).

**Figure 6 nutrients-12-01198-f006:**
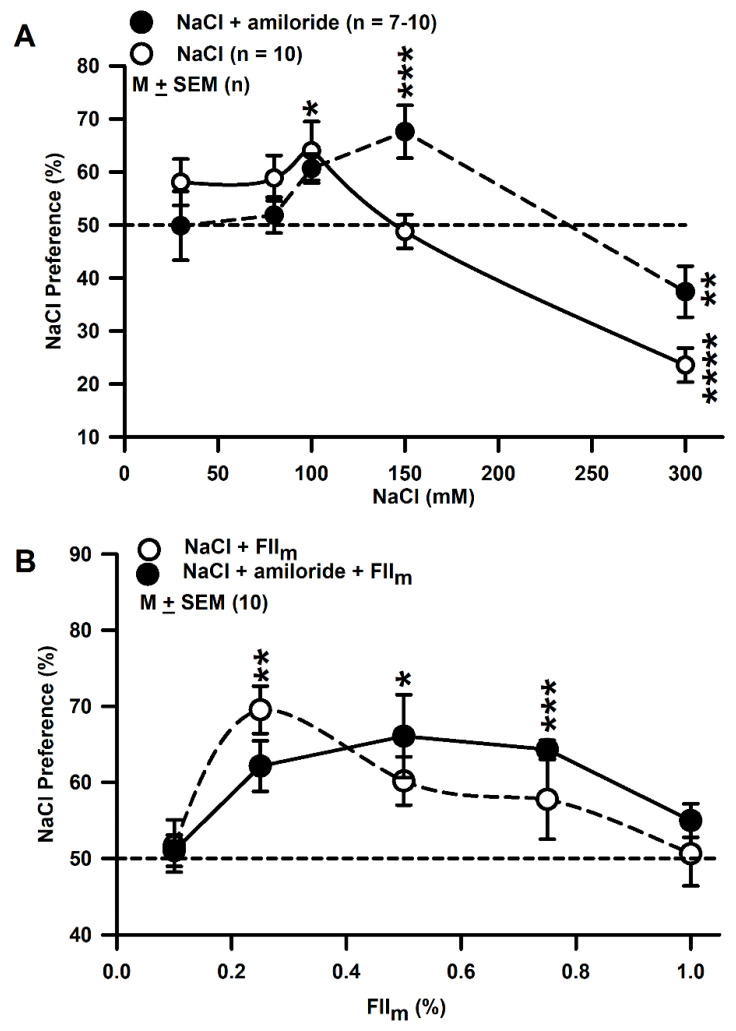
Effect of amiloride and FII_m_ on NaCl Preference in WT mice. (**A**) Shows NaCl Preference in WT mice when given a choice between H_2_O and varying concentrations of NaCl (3, 80, 100, 120, 150, 200 and 300 mM) in the absence (○) and presence of 10 µM amiloride (●). The values are presented as mean (M) ± SEM of *n*, where *n* = 7–10. * *p* = 0.02; ** *p* = 0.0134; *** *p* = 0.0024; **** *p* = 0.0001. (**B**) Shows NaCl Preference in WT mice when given a choice between H_2_O and 100 mM NaCl (○) or H_2_O and 100 mM NaCl + 10 µM amiloride (●) containing increasing concentrations of FII_m_ (0.1 to 1%). * *p* = 0.0086; ** *p* = 0.0018; *** *p* = 0.0001 (*n* = 10). Dotted line represents the indifference value.

**Figure 7 nutrients-12-01198-f007:**
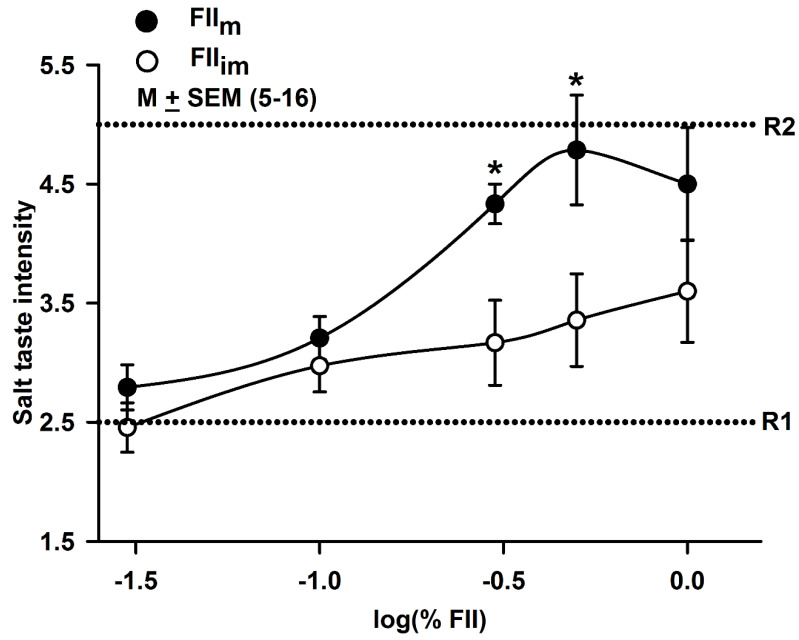
Effect of FII_m_ and FII_im_ on human salt taste intensity. Shows the effect of varying concentrations (0.03 and 1.0%) of FII_im_ (○) and FII_m_ (●) expressed in log units on human salt taste intensity. R1 corresponds to the intensity (2.5) of 0.2% NaCl and R2 corresponds to the intensity (5.0) of 0.35% NaCl. FII_m_ showed a significant (* *p* = 0.01) salt taste-enhancing activity at 0.003% and 0.005%. In contrast, no effect of FII_im_ was observed on human salt taste intensity over the concentration range between 0.03 and 1.0%.

**Figure 8 nutrients-12-01198-f008:**
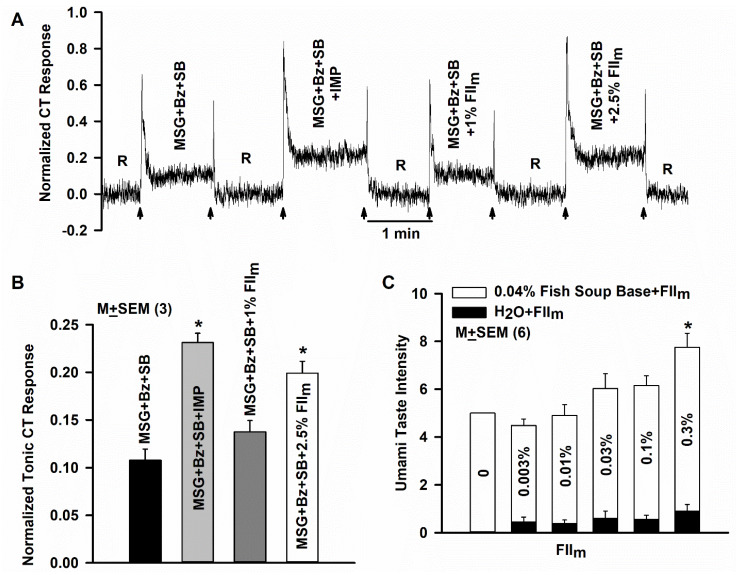
Effect of FII_m_ on the glutamate chorda tympani (CT) response and human umami taste sensory evaluation. (**A**) Shows a representative CT response in which the rat tongue was first rinsed with the rinse solution (R) and then with 100 mM MSG + 5 µM benzamil (Bz) + 1 µM SB-366791 (SB), MSG + Bz + SB +1 mM IMP, MSG + Bz + SB + 1% FII_m_ and MSG + Bz + SB + 2.5% FII_m_. The arrows represent the time period when the tongue was superfused with the rinse and stimulating solutions. (**B**) Shows mean normalized tonic CT responses from 3 rats. In each rat the data were normalized to the tonic response obtained with 0.3 M NH_4_Cl. * *p* = 0.001. (**C**) Shows the effect of adding increasing concentrations of FII_m_ (0.003 to 0.3%) to the 0.04% Fish Soup Base (open bars) or to H_2_O (filled bars). The values are presented as M ± SEM of n, where n represents the number of panel members tested. * *p* = 0.01.

**Table 1 nutrients-12-01198-t001:** Taste stimuli used for CT experiments.

	(mM)	Stimuli	(mM)
R	10 KCl + 10 HEPES	NaCl	10 KCl + 10 HEPES + 100 NaCl
R	10 KCl + 10 HEPES	NaCl + Bz	10 KCl + 10 HEPES + 100 NaCl + 0.005 Bz
R	10 KCl + 10 HEPES	NaCl + Bz + RTX	10 KCl + 10 HEPES + 100 NaCl + 0.005 Bz + RTX (0–0.01)
R	10 KCl + 10 HEPES	NaCl + Bz + FII or sub-fractions	10 KCl + 10 HEPES + 100 NaCl + 0.005 Bz + FII or sub-fractions
R	10 KCl + 10 HEPES	NaCl + SB	10 KCl + 10 HEPES + 100 NaCl + 0.001 SB
R	10 KCl + 10 HEPES	NaCl + SB + FII	10 KCl + 10 HEPES + 100 NaCl + 0.001 SB + FII or sub-fractions)
R	10 KCl + 10 HEPES	N + Bz + SB + FII	10 KCl + 10 HEPES + 100 NaCl + 0.005 Bz + 0.001 SB + FII or sub-fractions
R	10 KCl	MSG + Bz + SB	10 KCl + 100 MSG + 0.005 Bz + 0.001 SB
R	10 KCl	MSG + Bz + SB + IMP	10 KCl + 100 MSG + 0.005 Bz + 0.001 SB + 1 IMP
R	10 KCl	MSG + Bz + SB + FII	10 KCl + 100 MSG + 0.005 Bz + 0.001 SB + FII
R	10 KCl	Control-1	300 NH_4_Cl
R	10 KCl	Control-2	300 NaCl

4-(2-Hydroxyethyl)-1-piperazine ethanesulfonic acid (HEPES) was used to buffer the pH of rinse and salt stimuli at pH 6.1. Bz (benzamil); RTX (resiniferatoxin); SB (SB-366791, N-(3-methoxyphenyl)-4-chlorocinnamide). All compounds were obtained from Sigma-Aldrich.

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
