# Peer review of "Kokumi Taste Active Peptides Modulate Salt and Umami Taste"

_nutrients, 2020, doi:10.3390/nu12041198_

Round 1

Reviewer 1 Report

This study was undertaken to test the taste properties of peptide fractions isolated from Korean soy sauce, Ganjang.  These peptides are believed to elicit the enigmatic sensation named kokumi  The authors use a combination of electrophysiological recordings from taste nerves in rats, behavioral studies in mice, and human sensory evaluations to conclude that certain of the peptides enhance amiloride-insensitive salt and umami tastes.

This reviewer found it difficult to read and understand this ms and can suggest certain edits to help guide readers.

First, terminology and stimuli.  The authors refer to FII peptide fractions as "immature" and "matiure".  Unless these descriptors are the accepted jargon in this field, it might be more readable to name them "fresh" or "young" (as in cheeses or wines, respectively) versus "mature".  "Immature" seems an odd term and at first was confusing.  More important, it would be vital to know: (1) the Na+ concentration in the 0.1% to 5% FII isolates.  Could the observed enhancements/depressions simply be due to the additive effects of [Na] already present in the FII fractions?  This is probably unlikely, given that the authors statement (data not shown, however) that FII fractions alone do not stimulate chorda tympani nerve responses.  However, it would be comforting to know the total [Na] in their test stimuli; and (2) what is the concentration of FII fractions in Ganjang itself?  That is, do the concentrations of the FII stimuli used in this study (e.g., 0.1 -2.5%) at all reflect what is present in the real world.

Second, the rationale for including 3 different animal species was poorly presented.  Chorda tympani nerve recordings, RT PCR, and immunostaining were obtained from rats.  Two bottle taste behavioral assays were carried out on mice.  Sensory evaluations were tested by a human taste panel.  There are important differences in gustation among these 3 species and the justification for the multi-species approach was not explained well.  Indeed, the authors might want to consider publishing the behavioral data (mice, humans) in a separate manuscript..  This would make more sense to this reviewer,.  In any case, there was an insufficient summary of how the data from the 3 different studies (rats, mice, humans) were related.  As an aside, the authors recorded chorda tympani nerve responses only from female rats.  Is there some reason male rats were not included?  Are there differences in the effects of FII fractions in males versus females?  Male and female mice were tested in the 2 bottle taste preference assays.  Men and women were tested in the human sensory panel.  Were any differences between sexes observed?

Third, the presentation of the methods and data could be improved.  Specifically, (1) there is no reason to write out the sequences of stimulation (e.g., "R-> NH4Cl -> R" etc., lines 129-157).  This information is redundant; it is presented in the figures themselves.  And importantly, writing it out in text form like this is clumsy.  (2) The statement "CT responses were recorded under zero lingual current clamp..." (lines 110-111) is unnecessary information and potentially confusing; (3) Data listed in Table 1 seems extremely repetitive.  How important is it to list the stimuli in this fashion?  (4) The symbols in many of the figures (e.g. Fig 1B; 3C; 4A,B) are so large that they obscure the sem bars, if there are any.  The authors might want to change these symbols to reveal the error bars. (5) Quite importantly, the authors use outdated plotting (bar graphs).  Most top rank journals today require showing the actual data points (either alone or superimposed on the bar graphs) to reveal the data distributions, particularly for small sample sizes (e.g. N <10-20).  To be perfectly frank, this reviewer has difficulty believing that the sem of data in Fig. 2C are so small.  Were the results truly that uniform?

Fourth, the citation “[42]” in line 389 appears to be in error.  Also, the statement and citations in line 34 are incomplete.  Salty taste has been ascribed to all 3 taste cell types, not just Type I and Type III taste cells (e.g., Roebber et al, J Neurosci, 2019 ; Nomura et al, Neuron, 2020).

Fifth, the ms seems to lack a Discussion section where the data are interpreted and compared with other findings.  Perhaps one could argue that such discussion was included in the Results throughout the ms and perhaps this is the particular style for this journal.  But the short, one paragraph summary (lines 523-533) seemed an insufficient discussion of the findings that span 3 different species and several different types of experiments.

Finally, regarding the conclusion that FII peptide fractions influence salt and umami taste, it seems important to test (or at least acknowledge the lack of such a test for) other tastes (e.g. sweet, sour, etc).  For instance, could it be that FII somehow influences all tastes, not just salty and umami?  Indeed, the authors somewhat introduce this possibility vis-à-vis sour (acid) taste in lines 439-440.  If FII had such global effects on gustation (or for that matter, possibly even somatosensation, see lines 439-440), how might this influence/alter the interpretation of their findings?  Although the authors might argue such tests are beyond the scope of their study, it would seem important at least to acknowledge this lack of information about other tastes/sensations.

Author Response

We thank the reviewer for taking the time to read our paper carefully and providing detailed suggestions to improve the readability and understanding of the manuscript. Below we provide point by point responses to the reviewer’s comments:

  1. The reviewer states that the authors refer to FII peptide fractions as "immature" and "mature".  Unless these descriptors are the accepted jargon in this field, it might be more readable to name them "fresh" or "young" (as in cheeses or wines, respectively) versus "mature".  "Immature" seems an odd term and at first was confusing. 

Response:

It seems that while the reviewer is comfortable with the term mature, immature seems an odd and confusing term. We are ambivalent about this suggested change in the paper. Since, this seems to be just a readable issue, we request that immature designation for 1 year old FII fractions (FIIim) be retained in the paper.

  1. The reviewer states that, more important, it would be vital to know: (1) the Na+ concentration in the 0.1% to 5% FII isolates.  Could the observed enhancements/depressions simply be due to the additive effects of [Na] already present in the FII fractions?  This is probably unlikely, given that the authors statement (data not shown, however) that FII fractions alone do not stimulate chorda tympani nerve responses.  However, it would be comforting to know the total [Na] in their test stimuli; and (2) what is the concentration of FII fractions in Ganjang itself?  That is, do the concentrations of the FII stimuli used in this study (e.g., 0.1 -2.5%) at all reflect what is present in the real world.

Response:

We did not measure [Na] in FII fractions or FII fraction in Ganjang itself. However, we agree with the reviewer that it would be comforting to know the total [Na] in the test stimuli and the concentration of FII fractions in Ganjang itself. The reviewer’s point is well taken. Attempts to increase the concentration of FII in Ganjang could increase its effectiveness in enhancing salt taste. We hope that this paper may provide the incentive to the manufacturers of Ganjang to take this point of view.

In the revised version of the paper, the following statement has been added to the summary statement “The active component(s) and their salt enhancing property of naturally occurring MRPs by longer maturation in food should be further investigated for a better understanding of the potential link between the compound and its beneficial effect to reduce salt intake in human population [57, 58].”

           Similar to case with synthetic MRPs [Katsumata et al. 2008], adding FIIm to the rinse solution did not induced a CT response above baseline (data not shown). In addition to this, Fig. 1B shows that in the presence of 100 mM NaCl, FIIim over the whole range of concentrations used in the experiment did not induce a CT response above baseline. Fig. 1 also shows FIIm produces a biphasic response in the NaCl + Bz CT response. The maximum increase in the mean normalized tonic CT response occurred at 0.5% of FIIm, an 88% increase relative to NaCl + Bz tonic CT response in the absence of FIIm. Above 0.5%, the FIIm-induced increase in the NaCl + CT response decreased with increasing FIIm concentration. Most importantly, at 1% FIIm, the magnitude of the Bz-insensitive NaCl CT response was below the basal (i.e. in the absence of FIIm) NaCl + Bz CT response, and just above the rinse baseline (p = 0.0466; n = 3). Any addition of extra salt in the FIIm fraction at 1% concentration would be expected to increase the NaCl CT response. Taken together the results indicates that even at the highest concentration of FIIm or FIIim, the amount of Na+ present in the FII fractions does not contribute to the CT response.

  1. The reviewer states that the rationale for including 3 different animal species was poorly presented.  Chorda tympani nerve recordings, RT PCR, and immunostaining were obtained from rats.  Two bottle taste behavioral assays were carried out on mice.  Sensory evaluations were tested by a human taste panel.  There are important differences in gustation among these 3 species and the justification for the multi-species approach was not explained well. 

Response:

We agree. The reviewer’s point is well taken.

(i) Rationale for including 3 different animal species. In the revised version of the paper Introduction Section has been modified to present the rational for including 3 different animal species.

Studies performed by many labs, including ours, demonstrate that the amiloride- and benzamil (Bz)-insensitive salt taste receptors are the predominant transducers of salt taste in humans. Our previous studies demonstrate that over a range of concentrations, Xyl-MRPs reversibly enhanced the Bz-insensitive NaCl chorda tympani (CT) taste nerve response in rodents, whereas, at high concentrations, inhibited the Bz-insensitive NaCl CT response. The effect of Xyl-MRPs on the Bz-insensitive NaCl CT responses were TRPV1-dependent. In human sensory evaluation, at low salt concentrations, galacturonic acid MRPs (GalA-MRPs) enhanced human salt taste perception. These data suggest that in both rodents and humans, MRPs induce changes in amiloride-insensitive salt taste and umami taste.

In this paper, we investigated the effects of a naturally occurring MRPs fraction (500-10,000 Da, FII) isolated from immature (FIIim; 1-year old) and mature (FIIm; 4-year old) JGN on amiloride-insensitive salt taste responses and umami taste responses in humans and rodents. Effects of FIIm and FIIim were investigated on the Bz-insensitive NaCl CT responses and glutamate CT responses in rats and their interactions with TRPV1 modulators, and on behavioral responses to NaCl in C57BL/6 mice. To investigate if effects of FIIm on salty and umami taste responses are also observed in humans, we also investigated the effect of FIIm and FIIim on the sensory evaluation of salt and umami tastes in human subjects. Our results suggest that FIIm enhances amiloride-insensitive salt taste and umami taste.

(ii) CT Responses in rats and mice. Although the data was not included in the original version of the paper, we did make a limited number of recordings of the Bz-insensitive NaCl CT response in wild type (WT; C57BL/6J) and homozygous TRPV1 knockout mice (B6. 129S4-Trpv1tmijul; The Jackson Laboratory, Bar Harbor, ME). Consistent with our earlier study with MRPs (Katsumata et al. 2008), FIIm produced a similar biphasic response on the Bz-insensitive NaCl CT response in both SD rats and WT mice. Again, similar to the case with MRPs, in TRPV1 KO mice, FIIm (0.4%) did not induce CT response above the rinse baseline. In order to remain within the original number of figures, in the revised version of the paper, these results are presented as data not shown. This is akin to our results in rats. SB inhibited the basal Bz-insensitive NaCl CT response. In the continuous presence of SB, FIIm produced a significantly smaller increase in the Bz-insensitive NaCl CT response relative to in the absence SB (Fig. 2). TRPV1 KO mouse data is provided here only for the reviewer to see that we do indeed have limited data on WT and KO mice. These results demonstrate that MRPs (Katsumata et al. 2008) and FIIm produce similar effects on Bz-insensitive NaCl CT responses in both rats and mice.

(iii) Behavior studies in mice versus rats. As shown in our earlier study (Coleman et al. 2011), rats have a high preference for NaCl even in the presence of 5 mM Bz. Because of already high background NaCl preference, small increases in NaCl preference are difficult to evaluate. In contrast, WT mice demonstrate a more moderate preference for NaCl (revised Fig. 6) and small shifts in the preference curve are easily detected. For this reason, we choose to perform behavioral studies in mice.

In the revised version of the paper, this information has been included in section 2.3.

(iv) Human Sensory Evaluation. It is quite apparent that in human sensory evaluation of salt taste with FIIm, CT and behavioral data in rodents can help to interpret the data. Both humans and mice demonstrated biphasic effect on salt taste.  

  1. The reviewer states that indeed, the authors might want to consider publishing the behavioral data (mice, humans) in a separate manuscript.  This would make more sense to this reviewer.  In any case, there was an insufficient summary of how the data from the 3 different studies (rats, mice, humans) were related.  As an aside, the authors recorded chorda tympani nerve responses only from female rats.  Is there some reason male rats were not included?  Are there differences in the effects of FII fractions in males versus females?  Male and female mice were tested in the 2 bottle taste preference assays.  Men and women were tested in the human sensory panel.  Were any differences between sexes observed?

Response:

  1. As stated above, it is quite apparent that to in human sensory evaluation of salt taste with FIIm, CT and behavioral data in rodents can help to interpret the data. Both humans and mice demonstrated biphasic effect on salt taste. In our view, the strength of the paper lies in being able to connect neural recordings and behavior in rodents with human sensory evaluation. Most importantly, we demonstrate that this effect is primarily on the Bz-insensitive salt taste pathway that is TRPV1-dependent. It is important to mention that the other 2 reviewers did not raise this issue or concern.

  1. Our lab has been performing CT recordings for the past 20 years on both male and female rats. As reported earlier (Colemann et al. 2011), in our hands, because of bleeding issues during CT nerve isolation with male rats, we have a significantly greater success rate with CT recordings from female rats. In contrast, in both male and female mice the success rates is similar. We also chose female rats in this study so that we can compare the results obtained in this study with our earlier results with other amiloride-insensitive CT response modulators that were also obtained using female rats.
  1. The reviewer indicates that the presentation of the methods and data could be improved.  Specifically, (1) there is no reason to write out the sequences of stimulation (e.g., "R-> NH4Cl -> R" etc., lines 129-157).  This information is redundant; it is presented in the figures themselves.  And importantly, writing it out in text form like this is clumsy.

Response:

We agree. In the revised version of the paper text in lines 129-157 in the original version of the paper have been deleted.

  1. The reviewer comments that the statement "CT responses were recorded under zero lingual current clamp..." (lines 110-111) is unnecessary information and potentially confusing;

Response:

We agree. In the revised version of the paper this statement has been deleted.

  1. Data listed in Table 1 seems extremely repetitive. How important is it to list the stimuli in this fashion? 

Response:

With the deletion of sequences of stimulation (lines 129-157), Table 1 serves an important function, in that, the reader can quickly see the concentrations of stimuli and blockers used in each experiment without going into the text and figures. We request that Table 1 should be retained in the paper. 

  1. The symbols in many of the figures (e.g. Fig 1B; 3C; 4A,B) are so large that they obscure the sem bars, if there are any.  The authors might want to change these symbols to reveal the error bars. Quite importantly, the authors use outdated plotting (bar graphs).  Most top rank journals today require showing the actual data points (either alone or superimposed on the bar graphs) to reveal the data distributions, particularly for small sample sizes (e.g. N <10-20).  To be perfectly frank, this reviewer has difficulty believing that the sem of data in Fig. 2C are so small.  Were the results truly that uniform?

Response:

In both rats and mice the Bz-insensitive NaCl CT response is approximately 30% of the total NaCl CT response.

When presenting data in most figures we have provided examples of normalized CT tracing and the computed data from at least 3 rats. In our recordings, we only present data from animals in which CT responses to 0.3M NH4Cl before and after the experiment do not differ by more than 10% (see Fig. 5). Under these stringent conditions, within a stable CT recording, stimulating the tongue with a given concentration of NaCl repeatedly will give a mean normalized CT response with a very small error bar. Accordingly, in the case of peptide fractions where the dose-response relationship is shifted far to the right on the concentration axis, NaCl CT responses in the presence of the peptide are not different from the NaCl CT response in the absence of the peptide. Since we are comparing the magnitude of the NaCl CT response in the presence of the peptide with the basal NaCl CT response within the same animal, these differences, the means values and error bars are going to be small (see Fig. 3B).

In the case of FIIim, at all concentrations of FIIim tested, the NaCl + Bz CT response was not different from the basal NaCl CT response in each rat (Fig. 1B).

In the case of FIIm, the error bars are easily observed at 0.5 and 1% concentration. At these concentrations the maximum enhancement and inhibition of NaCl + CT response varies a bit from rat to rat.

In addition, the normalized values of CT responses to NaCl between animals are pretty close and the error bars of the mean data are small. This point is further illustrated by our observation that relative to control (NaCl + Bz), 0.5% FIIm (Fig. 1A) produced an equivalent maximum increase in the tonic NaCl + Bz CT response as 1 µM RTX (Fig. 4B) in different groups of rats. In our previous studies, most of the modulators of the Bz-insensitive NaCl CT response produce the maximum increase in the NaCl + Bz CT response by 80-100%, albeit at different concnetrations. In this study, FIIm increased NaCl + Bz CT response by 88%.

In another example tracing shown in comment 3 above, in TRPV1 KO mice the basal Bz-insensitive NaCl CT response in the absence and presence of FIIm is pretty much at the rinse baseline. Under these conditions the mean value and error bars between KO mice is very small and is barely visible in the graph.  

In addition to plotting mean ± SEM values, in all figures the data points are connected by smooth curves. The curves are generated using a fitting function that models the characteristic biphasic property of the peptide agonists of the amiloride-insensitive response. The biphasic property has been observed with every agonist of the amiloride-insensitive NaCl CT response thus far examined (Coleman et al. 2011; Dewis et al. 2013; Katsumata et al. 2008; Lyall et al. 2004, 2005, 2007; Treesukosol et al. 2007). Decreasing the size of the symbols will only make the distinction between different curves more obscure.

  1. The reviewer states that the citation “[42]” in line 389 appears to be in error.  Also, the statement and citations in line 34 are incomplete.  Salty taste has been ascribed to all 3 taste cell types, not just Type I and Type III taste cells (e.g., Roebber et al, J Neurosci, 2019 ; Nomura et al, Neuron, 2020).

Response:

Thanks.

  1. In the revised version of the paper, the statement and citations in line 34 have been modified. The new text now reads: Mammals use G-protein-coupled receptors (GPCRs) expressed in Type II taste receptor cells (TRCs) to detect bitter, sweet, and umami taste stimuli. While amiloride-sensitive salt taste is detected by Type 1 cells expressing epithelial Na+ channels, Type II and Type III cells mediate amiloride-insensitive salt taste. Otopetrin-1 proton selective channel expressed in Type III TRCs detect sour taste stimuli [1-3, 42, 43, 58].
  2. The text related to the citation “[42]” in line 389 in the original version of the paper has been deleted in the revised version of the paper.
  1. The reviewer states that the ms seems to lack a Discussion section where the data are interpreted and compared with other findings.  Perhaps one could argue that such discussion was included in the Results throughout the ms and perhaps this is the particular style for this journal.  But the short, one paragraph summary (lines 523-533) seemed an insufficient discussion of the findings that span 3 different species and several different types of experiments.

Response:

This may be due to the style of the journal, where Results and Discussion sections are combined. The Discussion of Results seems somewhat sparse.

In the revised version of the paper the Discussion section includes:

  1. Comparison of the effects of FIIm, a naturally occurring MRPs, with our previous study using synthetic MRPs (Katsumata et al. 2008).
  2. Biphasic nature of the CT response.
  3. Biphasic nature of the behavioral response in mice. Why mice were used for behavioral studies rather than rats.
  4. Biphasic nature of the human salt sensory evaluation
  5. TRPV1-dependence of Bz-insensitive NaCl CT responses in the absence and presence of FIIm.
  6. Lack of TRPV1 expression in TRCs
  7. Effect of CGRP on NaCl CT response. This may provide a novel relationship between the trigeminal system and salt taste
  8. RTX, a modulator of the Bz-insensitive NaCl CT response, at doses that maximally enhance or inhibit the CT response do not alter CT responses to sweet, bitter and sour stimuli
  9. Effect of FIIm on umami taste
  10. FIIm can induce an equivalent increase in the CT response to MSG as IMP
  11. The summary paragraph has been expanded to include the new information garnered from this study.
  1. The reviewer states that regarding the conclusion that FII peptide fractions influence salt and umami taste, it seems important to test (or at least acknowledge the lack of such a test for) other tastes (e.g. sweet, sour, etc).  For instance, could it be that FII somehow influences all tastes, not just salty and umami?  Indeed, the authors somewhat introduce this possibility vis-à-vis sour (acid) taste in lines 439-440.  If FII had such global effects on gustation (or for that matter, possibly even somatosensation, see lines 439-440), how might this influence/alter the interpretation of their findings?  Although the authors might argue such tests are beyond the scope of their study, it would seem important at least to acknowledge this lack of information about other tastes/sensations.

Response:

The reviewer raises a very point regarding the effect of FIIm on other tastes. Our results indicate that FIIm does not modulate Na+ detection via the amiloride-sensitive ENaCs. It specifically affects the Bz-insensitive Na+ detection via a non-specific cation pathway in a biphasic manner. At the concentrations that inhibit the amiloride-insensitive NaCl CT responses, it enhances umami taste responses. In our earlier study (Lyall et al. 2004), RTX demonstrated a biphasic response on the rat Bz-insensitive NaCl CT response. At 1 mM, it maximally enhanced and at 10 mM, maximally inhibited the Bz-insensitive NaCl CT response. At 1 and 10 mM concentrations, RTX did not alter CT responses to 500 mM sucrose, 10 mM quinine and 10 mM HCl. These results tend to suggest that over the concentration range that alter the Bz- insensitive NaCl CT response, modulators of the amiloride-insensitive pathway may not alter sweet, bitter or sour taste. At present, it is not known if FIIm concentrations alter taste responses to other taste stimuli.

In the revised version of the paper, this information is added to the discussion section.    

Reviewer 2 Report

General concerns:

  1. There are many abbreviations in this article. It is hard to find the full name for some of the abbreviations when they were first used, e.g., RTX. If the journal allows, it will be more author-friendly to provide a list of full names for the abbreviations somewhere, e.g., at the bottom of first page.
  2. The phenomenon that kokumi taste substances modulate salt and umami tastes has been reported. The authors need to make it more clearly what new information they are providing.
  3. Since amiloride-insensitive Na+ channels are expressed in taste bud cells in the posterior tongue that is innervated by the IXth cranial nerve, the authors need to clarify why the VIIth nerve was selected for this study.
  4. Rats and mice are different physiologically in many ways. The authors need to provide reasons why the behavioral tests were done in mice instead of rats that were used for the nerve recordings.

Specific concerns:

  1. In Figure 2, 0.4% FIIm was used, which is not the concentration (0.5%) with maximal effect shown in Figure 1. It will be good that the author tell the reason if there is one.
  2. In Figure 5, the authors showed the CGRPR expression in the circumvallate papilla that is innervated by the IXth nerve, not chorda tympani (VIIth). The images need to be replaced with the corresponding data from fungiform and/or anterior foliate papillae where taste buds are innervated by the VII nerve.
  3. It was described that FIIm was obtained as aromatic, basic, acidic, and neutral conjugated peptide fractions. The effects were presented in Figure 4 and section 3. 6.. It is not clear which fraction(s) was used in the results shown in other parts, e.g., Fig. 1-3.
  4. In Figure 6, the authors used i.p. injection of CGRP instead of oral administration. CGRP has a broad impact. It would be good that the authors discuss the specificity of the effects of CGRP.
  5. In Figure 4 and 8, the concentrations of the reagents were presented as log value. It needs to be pointed out or consistent with the descriptions in the text of Results.

Author Response

We thank the reviewer for carefully reading our paper and for his/her fair and helpful comments to improve the presentation of results and the interpretation of the data. Below we provide a point by point response to reviewer’s comments:

  1. There are many abbreviations in this article. It is hard to find the full name for some of the abbreviations when they were first used, e.g., RTX. If the journal allows, it will be more author-friendly to provide a list of full names for the abbreviations somewhere, e.g., at the bottom of first page.

Response:

We agree. In the revised version of the paper all abbreviations are defined in parentheses the first time they appear in the abstract, main text, and particularly in figures and Table captions and used consistently thereafter.

  1. The phenomenon that kokumi taste substances modulate salt and umami tastes has been reported. The authors need to make it more clearly what new information they are providing.

Response:

We followed the reviewer’s suggestion and have made it more clear what new information is provided in this paper.

In our previous study (Katsumata et al. 2008), we artificially generated Maillard Reacted Peptides (MRPs) by conjugating a peptide fraction (1000 - 5000 Da) isolated from soy protein hydrolysate with different sugar moieties. We demonstrated that these artificial MRPs produce biphasic effects on the Bz-insensitive NaCl CT response in rodents and also produced biphasic effects on human salt taste perception. In this paper we present new data that demonstrate that a naturally occurring MRPs fraction (500-10,000 Da, FII) present in mature (FIIm; 4-year old) JGN also produces biphasic effects on the Bz-insensitive NaCl CT responses in rodents and also produces biphasic effects on human salt taste perception. These results demonstrate that naturally occurring MRPs are present in food or are generated during maturation and cooking process that can have salt modulating effects in humans.

       At present the underlying cellular mechanism by which MRPs and MRPs modulate human salt taste is not known. Here, we present new data that show that like MRPs, naturally occurring FIIm fraction modulates the amiloride- and Bz-insensitive NaCl CT response in rodents in a biphasic manner. Bz-insensitive NaCl CT response in the absence and presence of FIIm were sensitive to SB and were not observed in TRPV1 KO mice. In addition, in the presence of a sub-threshold concentration of RTX, FIIm failed to enhance Bz-insensitive NaCl CT response. Based on these results we hypothesize that FIIm effects on Bz-insensitive NaCl CT response are TRPV1 dependent. Since rodent TRCs do not seem to express TRPV1, we provide new data that suggests that FIIm and other modulators can indirectly alter CT responses to NaCl via the release of CGRP from trigeminal fibers near the fungiform taste buds in the anterior taste field. These new data may present a novel relationship between trigeminal system and salt taste perception.  

          The last summary paragraph has been modified to incorporate this new information.    

  1. Since amiloride-insensitive Na+ channels are expressed in taste bud cells in the posterior tongue that is innervated by the IXth cranial nerve, the authors need to clarify why the VIIth nerve was selected for this study.

Response:

The reviewer is correct that the glossopharyngeal nerve responses to NaCl are completely amiloride insensitive and the circumvallate taste filed seems to lack functional ENaCs. However, it is important to emphasize that although the predominant NaCl CT response in rodents is amiloride sensitive (~70%), a significant part of the NaCl CT response is Bz- and amiloride insensitive across the concentration-response range of NaCl (DeSimone et al. 2001).

       In previous studies from our lab, we have investigate the effect of various modulators (RTX, CAP, MRPs, NGCC, cetylpyridinium chloride (CPC), nicotine, alcohol, PIP2, IP3, and Ca2+) on the Bz-insensitive NaCl CT response using both rats and mice. To compare the results of the effects of FIIm on the Bz-insensitive NaCl CT response with previous published results with other modulators, in this study we stayed with CT recordings in rats and mice.

       Most importantly, the identity of the amiloride-insensitive receptor at present, at best, remains elusive. Our previous studies using CT recordings under lingual voltage-clamp indicate that in the anterior tongue the amiloride-insensitive receptor it is a non-selective cation channel (Lyall et al. 2004). Our current and previous results indicate that the channel is sensitive to SB, capsazepine (CZP), iodo-RTX (I-RTX), temperature, and the modulators listed in the paragraph above.

In the revised version of the paper, this information has been incorporated in section 2.2

  1. Rats and mice are different physiologically in many ways. The authors need to provide reasons why the behavioral tests were done in mice instead of rats that were used for the nerve recordings.

Response:

             Although the data was not included in the original version of the paper, we did make a limited number of recordings of the Bz-insensitive NaCl CT response in wild type (WT; C57BL/6J) and homozygous TRPV1 knockout mice (B6. 129S4-Trpv1tmijul; The Jackson Laboratory, Bar Harbor, ME). Consistent with our earlier study with MRPs (Katsumata et al. 2008), FIIm produced a similar biphasic response on the Bz-insensitive NaCl CT response in both SD rats and WT mice. Again, similar to the case with MRPs, in TRPV1 KO mice, FIIm (0.4%) did not induce CT response above the rinse baseline. In order to remain within the original number of figures, in the revised version of the paper, these results are presented as data not shown. This is akin to our results in rats. SB inhibited the basal Bz-insensitive NaCl CT response. In the continuous presence of SB, FIIm produced a significantly smaller increase in the Bz-insensitive NaCl CT response relative to in the absence SB (Fig. 2). TRPV1 KO mouse data is provided here for the reviewer to see that we do indeed have limited data on WT and KO mice.

As shown in our earlier study (Coleman et al. 2011), rats have a high preference for NaCl even in the presence of 5 mM Bz. Because of already high background NaCl preference, small increases in NaCl preference are difficult to see. In contrast, WT mice demonstrate a more moderate preference for NaCl (revised Fig. 6) and small shifts in the preference curve are easily detected.

In the revised version of the paper, this information has been included in section 2.3.  

Specific concerns:

  1. In Figure 2, 0.4% FIIm was used, which is not the concentration (0.5%) with maximal effect shown in Figure 1. It will be good that the author tell the reason if there is one.

Response:

The reviewer is correct in pointing out this discrepancy. In generating Fig. 1B, we used FIIm concentrations of 0.001, 0.025, 0.1, 0.2, 0.25, 0.5, 0.75, and 1 percent. We did not use 0.4% FIIm. However, calculating from the fitted curve in Fig. 1B, the mean increase in Bz-insensitive NaCl CT response at 0.4% FIIm was not different from its value at 0.5% FIIm (0.1863 vs 0.1865). This is because the fitted curve is flat at FIIm concentrations around 5%.

In the revised version of the paper the following text has been added: Because 0.4% and 0.5% FIIm give almost equivalent CT responses (Fig. 1B), we used 0.4% FIIm in these experiments.

  1. In Figure 5, the authors showed the CGRPR expression in the circumvallate papilla that is innervated by the IXth nerve, not chorda tympani (VIIth). The images need to be replaced with the corresponding data from fungiform and/or anterior foliate papillae where taste buds are innervated by the VII nerve.

Response:

We agree with the reviewer that demonstrating CGRPR expression in fungiform taste buds cells would be consistent with CT data presented in this paper. However, currently, we do not have immunostaining of CGRPR in fungiform taste bud cells. Since, CGRPR expression has been demonstrated in mice circumvallate taste cells, this data (original Fig. 5) and section 3.7 have been deleted from the revised version of the paper. Since CGRP effects were observed on the NaCl CT response (original Fig. 6, now revised Fig. 5), it is most likely that CGRPR expression is present in a subset of fungiform taste bud cells. The lack of this data in the fungiform taste receptive field has been acknowledged in the revised version of the paper.  

  • It was described that FIIm was obtained as aromatic, basic, acidic, and neutral conjugated peptide fractions. The effects were presented in Figure 4 and section 3. 6.. It is not clear which fraction(s) was used in the results shown in other parts, e.g., Fig. 1-3.

Response:

Data presented in Figs. 1-3 was obtained with the original un-fractionated FIIim and FIIm. The fractions were designated as FIIm(a-d). In the revised version of the paper, this distinction has been emphasized in section 2.1.

  1. In Figure 6, the authors used i.p. injection of CGRP instead of oral administration. CGRP has a broad impact. It would be good that the authors discuss the specificity of the effects of CGRP.

Response:

In our immuno pictures of circumvallate taste bud cells, we did not see specific labelling at the apical tips (this data has been deleted from the revised version). This tends to suggest that CGRPR may be predominantly located at the basolateral membrane of cells. α-CGRP is a 37-amino acid neuropeptide. We were concerned that topical lingual application of CGRP may not be able to reach its target in the basolateral membrane of TRCs. Based on these considerations, we opted for intraperitoneal injection route. In the revised version of the paper, the following new text has been added: Due to the concern that topical lingual application of CGRP, a large neuropeptide, may not be able to reach its receptor in TRCs, CGRP was administered by intraperitoneal injection

  1. In Figure 4 and 8, the concentrations of the reagents were presented as log value. It needs to be pointed out or consistent with the descriptions in the text of Results.

Response:

             Since the concentration of FIIm varied from 0.001 to 1 percent (1000 fold), we choose to plot the data as log FIIm concentration. This point has been clarified in the figure legends to Figs. 1, 3, 4 and 8 (now figure 7).

Reviewer 3 Report

In this article, the authors investigated the effects of Maillard Reaction Peptides on salt and umami taste response in rodents and humans. The authors found the MRP from mature Ganjang to enhance salt and umami taste responses. Overall, the manuscript is well written with sufficient details for understanding the context and the significance of the work. The methods and results were clearly presented. For the human subject experiments, number of participants included in the study was small. However, authors could find some effect of the kokumi taste active peptide fraction on salt and umami taste. I do not have any major comments. The authors should check the p values reported in the paper. For example, for figure 2, the authors report “**p=0.0015”. Since there were multiple comparisons with p<0.01, it is unlikely for all p values to be the same.

Author Response

We thank the reviewer for recognizing that overall the manuscript is well written with sufficient details for understanding the context and the significance of the work. The methods and results were clearly presented. However, the reviewer did raise a concern regarding statistical analyses in Fig. 2C.

  1. The reviewer recommends that authors should check the p values reported in the paper. For example, for figure 2, the authors report “**p=0.0015”. Since there were multiple comparisons with p<0.01, it is unlikely for all p values to be the same.

Response:

             With reference to Fig. 2C, in the revised version of the paper, we now make clear in the figure legend that all unpaired comparisons were made with respect to the normalized value of the tonic CT response to NaCl + Bz at 23oC. The p values were recalculated. The p value for the comparison between the normalized value of the tonic CT response to NaCl + Bz at 23oC and 42oC was 0.0038 (*). The rest of the p values wrt the tonic CT response to NaCl + Bz at 23oC were indeed 0.0001 (**). This information has been corrected in the text.
